# Nanoscale Imaging of Biological Tissues: Techniques, Challenges and Emerging Frontiers

**DOI:** 10.3390/nano15231752

**Published:** 2025-11-22

**Authors:** Rohit Kajla, Rebecca Leija-Cardenas, Meghraj Magadi Shivalingaiah, Muhammad Waqas Shabbir, Zihao Ou

**Affiliations:** 1Department of Physics, The University of Texas at Dallas, Richardson, TX 75080, USA; 2Department of Electrical and Computer Engineering, The University of Texas at Dallas, Richardson, TX 75080, USA

**Keywords:** nanoscience, biomedical imaging, electron microscopy, super-resolution microscopy, expansion microscopy, atomic force microscopy, mass spectroscopy, artificial intelligence

## Abstract

Nanoscale characterization of biological tissues bridges molecular identity with structural, mechanical, and chemical organization, enabling high-resolution insights into intact specimens. This review provides a comprehensive overview of the principal imaging modalities that resolve cellular and subcellular features in biological tissues. Electron microscopy techniques offer ultrastructural details and volumetric reconstructions with sectioning and tomography techniques. Optical nanoscopy approaches such as single-molecule localization microscopy, stimulated emission depletion microscopy, structural illumination microscopy, and expansion microscopy achieve fluorescence-based mapping with tens-of-nanometer precision. Complementary platforms like atomic force microscopy and nanoscale secondary ion mass spectrometry extend nanoscale characterization into mechanical and chemical domains. Artificial intelligence has emerged as a transformative tool for segmentation, image restoration, and volumetric reconstruction, addressing bottlenecks in throughput and interpretability. From practical applications on biological tissues, we evaluate each technique’s strengths, limitations, and potential for clinical applications. The review concludes with a discussion on emerging directions, including live-tissue nanoscopy, correlative light and electron microscopy, and machine-driven high-throughput imaging for further investigation of nanoscale biological structures and functions.

## 1. Introduction

Biological tissues are hierarchically organized: Molecules assemble into macromolecular complexes, which form organelles, cells, and ultimately multicellular architectures [1]. Many of the most critical biological functions such as ion channel organization in neurons, collagen fibril alignment in dermis, mitochondrial cristae remodeling in muscle, or lipid and mineral distributions in bone are set at the 10–500 nm scale [2,3,4,5]. At this nanoscale, biological processes are tightly linked to mechanical, chemical, and electrical properties that underpin physiology and pathology. Yet, resolving, quantifying, and correlating these features across intact tissues remains a major challenge, because no single imaging modality simultaneously provides molecular specificity, nanometer resolution, large-volume coverage, and gentle preservation of native states [6,7,8,9,10,11,12,13,14].

The progress of nanoscale characterization techniques has always been linked to technological innovation [6,15,16,17,18]. Advances in cryo-preservation and tissue clearing strategies now enable visualization of increasingly native and complex samples [19,20,21,22,23,24,25]. Likewise, adaptive-optics-based designs extend imaging depth and contrast in thick specimens [26,27]. Continued development of instrumentation, detection systems, and sample-handling chemistries is essential to overcome persistent trade-offs between resolution, volume, and preservation of native structure [6,20,21,22,23,24,25,28,29]. Equally crucial are breakthroughs in computational analysis, automation, and artificial intelligence, which convert raw image data into quantitative, multiscale biological knowledge [30,31,32]. Each leap in imaging capability, from the first electron microscopes to modern super-resolution and cryogenic methods has redefined the biological scales that can be observed and quantified [33,34].

This review discusses the nanoscale characterization techniques for biological tissues, organized into four major topics (Figure 1). The first one is electron microscopy, which remains the gold standard for nanoscale imaging. The techniques discussed include scanning electron microscopy (SEM), transmission electron microscopy (TEM), scanning transmission electron microscopy (STEM), environmental scanning electron microscopy (ESEM), cryo-electron microscopy (cryo-EM), cryo-electron tomography (cryo-ET), and volume electron microscopy (vEM). The second topic is optical nanoscopy, which includes single-molecule localization microscopy (SMLM), stimulated emission depletion microscopy (STED), structured illumination microscopy (SIM), and expansion microscopy (ExM). The third topic is complementary modalities that probe properties beyond morphology, offering mechanical or chemical contrast, including atomic force microscopy (AFM) and nanoscale secondary ion mass spectrometry (nanoSIMS) and tip-based Raman spectroscopy (TERS). Lastly, we discuss the importance of artificial intelligence (AI) in bioimage analysis, which is essential in nanoscale characterizations.

This review is intended for both experimentalists and computational scientists who require a practical, side-by-side understanding of what each method offers, how to prepare and handle tissue samples, which pitfalls to avoid, and how modern AI pipelines transform raw image stacks into quantitative insights. For each technique, we start with a concise conceptual and technical overview and pair with experimental demonstrations. Across all four areas, we emphasize practical considerations for tissue studies: the achievable resolution, the condition of the specimen (live, fixed, cleared, or cryo-preserved), the labeling or contrast mechanism required, the imaging volume and throughput, and potential sources of artifacts or error. In this review, we focus on real-space direct imaging methods and scattering techniques, such as small angle x-ray and neutron scattering, which probe solution-averaged structures of biomolecules and complexes, are not included [35]. Finally, we do not attempt to provide an exhaustive catalogue of all nanoscopy modalities; instead, we highlight widely adopted and mature approaches that are rapidly gaining attention in biological research.

## 2. Electron Microscopy

Electron microscopy (EM) is a direct imaging technique for visualizing structures at the nanometer scale and resolving features smaller than the diffraction limit of conventional light microscopy [36]. Multiple EM modalities have been developed, each optimized to highlight complementary aspects of cellular and tissue organization. Scanning electron microscopy (SEM) provides detailed views of surface topography, while transmission electron microscopy (TEM) and scanning transmission electron microscopy (STEM) can resolve intracellular architecture with near-atomic precision [37]. Cryogenic approaches, including cryo-electron microscopy (cryo-EM) and cryo-electron tomography (cryo-ET), preserve hydrated native states and allow molecular complexes to be studied at high resolution [19,38]. In parallel, volume EM (vEM) techniques, such as serial block-face SEM and focused ion beam SEM, have extended imaging to large-scale three-dimensional reconstructions of cells and tissues [11,39,40]. These developments have transformed EM from a two-dimensional imaging modality into a platform for producing quantitative, three-dimensional maps of biological architecture, bridging scales from individual molecules to entire tissues [7].

### 2.1. Scanning Electron Microscopy

Scanning electron microscopy (SEM) is a technique originally developed to study the surface topography of materials [37]. The basic principle involves using a highly focused primary electron beam that is scanned across the specimen surface shown in Figure 2a, generating signals such as secondary electrons, which provide high-resolution morphological information, and backscattered electrons, which yield compositional and density contrast. Modern high-resolution SEM systems, particularly those using field emission sources, can achieve a resolution of less than 1 nm, well beyond the diffraction-limited resolution of light microscopy [41]. A major advantage of SEM is its ability to image large areas and volumes, and unlike TEM, it is not fundamentally limited by sample thickness. A limitation remains that achieving high-quality images often involves complex preparatory steps, such as gold coating or hexamethyldisilazane drying, which can require several hours before final imaging [42], and preparation methods play an essential role in the imaging contrast of SEM. As shown in Figure 2b, three approaches are compared side by side: conventional SEM with gold sputter coating, TEM with negative staining, and a rapid SEM method using ionic liquid infiltration on pre-coated filters [43]. Conventional sputter-coated SEM provides surface detail but introduces preparation artifacts such as wrinkling and 10–20% shrinkage of cells, while negative-stain TEM requires extensive chemical processing that can distort hydrated states. By contrast, the ionic liquid method is fast and can be completed in approximately 15 min and enables the imaging of unfixed, fully hydrated specimens. Together, these results emphasize how improved SEM preparation strategies can significantly enhance biological relevance by reducing distortion of artifacts.

Alongside improvements in imaging contrast, SEM has increasingly been applied in diagnostic and clinical contexts. High-resolution SEM with backscattered electron detectors has been utilized to examine resin-embedded tissue sections and enable visualization of multiple specimen types, including renal, lung, prostate, and brain tissues [44]. In clinical practices, SEM has been used to analyze biomaterials tissue interfaces and nanoscale morphologies of cells [45,46]. In one example, with the unique capabilities in resolving nanoscale surface morphology, SEM has provided critical insights into circulating tumor cells and tumor-derived extracellular vesicles [47]. By connecting nanoscale features with pathological data, SEM shows growing importance in diagnostic pathology and cell biology [42].

### 2.2. Transmission Electron Microscopy

Transmission electron microscopy (TEM) works by passing a beam of electrons through an ultrathin specimen, generating contrast as electrons are scattered or absorbed [36], as shown in Figure 2a. Preparation of biological samples for TEM imaging typically involves chemical fixation, embedding in epoxy resins such as Araldite or Durcupan, and cutting sections to thickness of 50–100 nm [48]. Heavy metal stains, including osmium tetroxide, uranyl acetate, and lead citrate, are then applied to enhance contrast, making structures like membranes appear darker [49]. With sub-nanometer resolution, TEM is one of the most powerful approaches for investigating the internal subcellular features. By imaging sections as thin as ~100 nm, TEM can reveal sharp details of structures such as sub-pellicular microtubules and flagellar components in protozoans like *Trypanosoma brucei* (Figure 2c) [50]. At the same time, the reliance on ultrathin sections introduces a fundamental limitation: three-dimensional organization is reduced to a two-dimensional section. As a result, larger organelles or extended structures are only partially represented, and spatial relationships between components can be difficult to interpret [51].

High resolution has made TEM a unique tool for studying membranes, cytoskeletal filaments, organelles, and for detecting subtle structural changes associated with disease. In clinical diagnostics, TEM has been used as one of a combination of tests to detect ciliary ultrastructure defects and other subcellular changes that characterize certain diseases [52]. TEM has also become indispensable in the development of nanomedicine, offering precise insights into nanoparticle uptake, localization, and biodistribution within cells and tissues [53]. Beyond simple visualization, TEM can be paired with stereological approaches to quantify particle distribution and methods such as the relative labeling index enable comparisons of particle density across cellular compartments to assess whether localization is random or compartment-specific [54].

### 2.3. Scanning Transmission Electron Microscopy

Scanning transmission electron microscopy (STEM) works by focusing a very fine electron probe onto the sample and scanning it across in a line-by-line raster pattern [55]. At each point, scattered electrons are collected by different detectors: bright-field detectors capture low-angle scattering, while dark-field and high-angle annular dark-field detectors capture higher-angle scattering. One of the key advantages of STEM for biological imaging is its ability to accommodate much thicker resin sections up to ~500 nm in a single acquisition [50]. This preserves continuity across larger volumes and makes it possible to view organelles in their entirety rather than as fragmented slices. In *Trypanosoma brucei*, for example, STEM can capture the kinetoplast, basal bodies, and extended portions of the flagellum within one field of view, providing a more complete structural context than conventional TEM thin sections (Figure 2c) [50]. This depth capability improves interpretation and enables more accurate quantitative measurements of distances and dimensions. Heavy atom contrasting agents further enhance STEM images of thick sections, giving particularly high contrast and a more homogeneous background than in TEM as shown in Figure 2c.

In addition to visualizing nanostructures, STEM can map elemental composition in biological samples using techniques such as Energy-Dispersive X-ray Spectroscopy, for example, Zinc in metalloproteins [56,57]. The introduction of advanced support materials, particularly graphene substrates, has further improved STEM studies by reducing background signal from conventional carbon films, thereby enhancing contrast and signal-to-noise ratios (SNR) for small biological masses [58,59].

Environmental scanning electron microscopy (ESEM) represents an extension of conventional SEM that enables imaging of biological tissues under partially hydrated, low-vacuum, or variable-pressure conditions. Unlike high-vacuum SEM, which requires extensive fixation, dehydration, and conductive coating, ESEM allows imaging of uncoated and minimally processed specimens by introducing water vapor or inert gas into the chamber to stabilize surface charge and prevent sample desiccation [60,61,62]. This capability is useful for soft tissues, cells, and extracellular matrix components that are easily distorted during drying steps. The secondary electron detector used in ESEM allows high-contrast imaging even at chamber pressures of several hundred Pa, which helps in the visualization of surface morphology, hydration-dependent features, and dynamic processes such as swelling, deformation, or phase transitions. In biological tissue research, ESEM has been used to observe hydrated cartilage, skin, plant tissues, biofilms, and wet cellular surfaces, providing micro- to nanoscale insights without conductive coating or high-vacuum artifacts [46,63,64,65,66]. While ESEM generally offers lower resolution than high-vacuum SEM, its ability to image delicate biological samples in near-native states makes it an essential complementary modality within the EM toolkit for nanoscale tissue characterization.

**Figure 2 nanomaterials-15-01752-f002:**
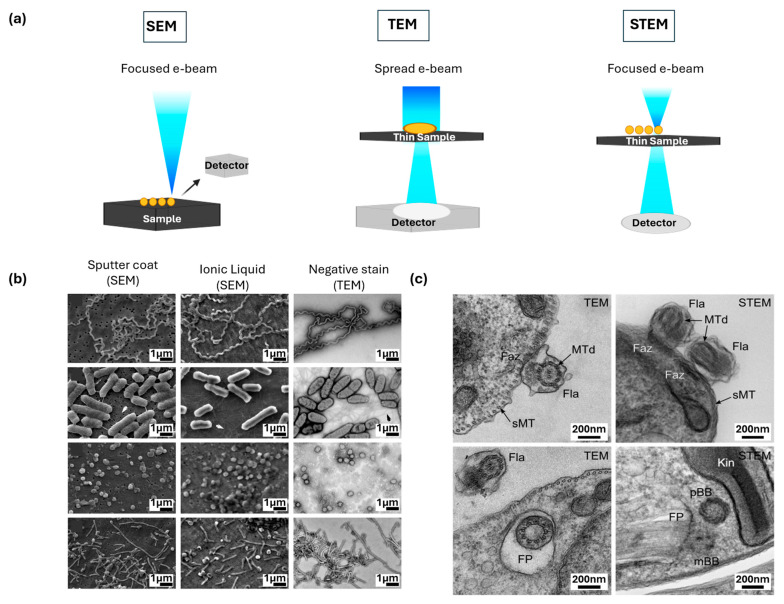
Scanning and transmission electron microscopy techniques for nanoscale characterization of biological tissues. (**a**) Schematics of SEM, TEM, and STEM, highlighting differences in electron beam configuration and signal detection. (**b**) Examples of SEM imaging, comparing sputter-coated, ionic liquid, and TEM negative staining approaches illustrated for different biological samples. Adapted with permission from Ref. [43]. Copyright 2016 Springer Nature. (**c**) TEM and STEM views of Trypanosoma brucei, capturing both extracellular flagellum structures and intracellular organization Adapted with permission from Ref. [50]. Copyright 2023 John Wiley & Sons, Ltd.

### 2.4. Cryo-Electron Microscopy

Cryo-electron microscopy (cryo-EM) enables high-resolution imaging of biological material preserved in a vitrified, frozen-hydrated state, which prevents ice crystal formation and preserves native ultrastructure [67]. Cryo-EM has transformed structural biology by enabling atomic-scale visualization of macromolecules often inaccessible to crystallography or NMR, including therapeutically important membrane proteins [18,38]. In the single-particle analysis workflow shown in Figure 3a, thousands of low dose 2D projection images of identical macromolecules are recorded at single tilt angles. These images are then computationally aligned, classified, and averaged to generate a three-dimensional reconstruction of the biomolecule [19,68]. Despite its success, cryo-EM faces several technical challenges: First, biological specimens interact only weakly with electrons and are highly sensitive to radiation damage, which restricts the allowable electron dose and results in images with inherently low signal-to-noise ratios [69,70,71]. Additionally, sample preparation remains a critical bottleneck: producing thin, uniform layers of vitreous ice is technically demanding, and artifacts such as particle adsorption to the air–water interface can distort molecular orientation and reduce the quality of reconstructions [72].

Cryo-EM has become a crucial imaging technique for both fundamental biology and drug discovery. In one example, cryo-EM has been successfully applied to clarify the assemblies of chronic wasting disease prion fibrils, which were reconstructed to 2.8 Å resolution (Figure 3c), providing molecular insight into mechanisms of disease transmissibility [73]. Two-dimensional images of vitrified fibrils revealed both individual and bundled morphologies, and single-particle images were combined to generate three-dimensional density maps that revealed a left-handed twist, completing a 180° rotation over a 295 nm cross-over distance. Beyond structural discoveries, computational advances have further accelerated cryo-EM characterization and analysis workflows. Deep learning based tools now automate particle picking, denoising, classification, and refinement, reducing processing times and improving reconstruction quality [74] and address long-standing issues like preferred orientation by generating auxiliary particles [75].

### 2.5. Cryo-Electron Tomography

Cryo-electron tomography (cryo-ET) is a 3D imaging technique designed to visualize the complex nanoscale organization of cells and tissues, preserving them in a vitrified state [76]. The technique as shown in Figure 3a involves acquiring a tilt series (a sequence of 2D projection images as the sample is rotated, typically ±70°), which are computationally reconstructed to form a 3D volume or tomogram [68]. For thick biological samples, preparation requires sophisticated micromachining techniques like cryo-focused ion beam milling to reduce the specimen thickness to the necessary 200–500 nm range for electron penetration [77]. While Cryo-ET achieves nanometer-scale resolution in 3D, it faces two practical limitations [78]: First is the “missing wedge” artifact, resulting from the incomplete tilt angle range, which distorts the resolution along the *z*-axis; Second is the high cumulative electron dose necessary for tilt series acquisition remains a major challenge.

Cryo-ET provides unique strength in connecting molecular detail with cellular context, offering insights into how nucleosomes assemble into higher-order chromatin architecture that cannot be obtained by single-particle analysis in cryo-EM alone. In one application, cryo-ET have revealed how chromatin is organized within intact human T-lymphoblast cell nuclei, capturing both the nuclear context and the arrangement of nucleosomes in situ (Figure 3b) [79]. Tomographic reconstructions made it possible to visualize chromatin fibers directly, while sub-tomogram averaging of nearly 7000 nucleosomes improved the resolution to 12 Å, sufficient to resolve DNA dyads, linker DNA, and the binding site of histone H1. Computational refinement via sub-tomogram averaging is essential for boosting signal and resolving repeated molecular complexes across tomograms [80]. To further improve interpretation, molecular labeling strategies are being developed like gold nanoparticles as cryo-ET tags to help identify specific proteins in the dense cellular background [81]. Using on-grid immunogold labeling prior to cryo-ET, Photosystem II complexes can be located within thylakoid membranes [82]. These advances further expand cryo-ET to combine molecular identification with high-resolution structural characterization.

**Figure 3 nanomaterials-15-01752-f003:**
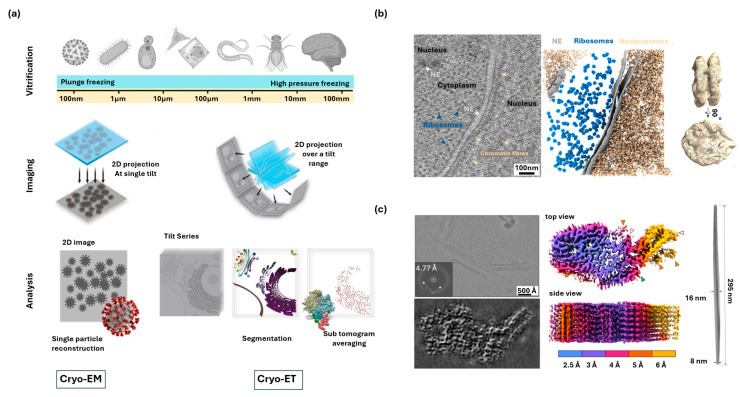
Cryo-EM and Cryo-ET for nanoscale analysis of biological tissues. (**a**) Overview of cryo-EM and cryo-ET workflows, illustrating vitrification, data collection, and reconstruction analysis strategies. Adapted with permission from Ref. [68]. Copyright 2025 Elsevier Ltd. (**b**) Cryo-ET of chromatin fibers, with tomographic slices, segmented nucleosomes and ribosomes, and sub tomogram averages fitted with atomic models. Adapted with permission from Ref. [79]. Copyright 2023 Springer Nature. (**c**) Cryo-EM of chronic wasting disease prion fibrils, showing raw images, density maps, and surface reconstructions colored by local resolution. Orange arrowheads: peripheral densities not attributed to polypeptide in the subsequent modeling; open arrowheads: the sites of potential N-linked glycans; blue arrowhead: C-terminus where GPI anchor is attached. Adapted with permission from Ref. [73]. Copyright 2024 Springer Nature.

### 2.6. Volume Electron Microscopy

Volume electron microscopy (vEM) encompasses a suite of automated imaging methods developed to generate large three-dimensional datasets of resin-embedded biological tissues at nanometer resolution, often thicker than a micron [83]. Traditional serial-section TEM (ssTEM) relies on labor-intensive manual cutting and alignment, and recent development of vEM integrates sectioning and imaging in automated workflows, enabling efficient reconstruction of extended tissue volumes. Here, we discuss several key strategies: Serial block-face SEM (SBF-SEM), which uses an in-chamber microtome to iteratively cut and image block faces; Focused ion beam SEM (FIB-SEM), which employs an ion beam to mill the block surface at steps as fine as 5 nm; Automated tape-collecting ultramicrotomy SEM (ATUM-SEM), which gathers ultrathin sections on tape for high-throughput imaging [84]. Figure 4a provides a schematic overview of several vEM techniques, including ssTEM, ET, SBF-SEM, FIB-SEM, and array tomography (AT), highlighting their differences in volume range and achievable resolution [85]. All these methodologies share a common workflow where specimens undergo heavy-metal staining and resin embedding, followed by automated imaging of incremental z-planes. Complex labor-intensive preparation, including heavy-metal staining protocols such as reduced Osmium–Thiocarbohydrazide–Osmium (rOTO), enhances membrane and protein contrast by sequential osmium deposition [86]. The resulting raw datasets are then post-processed through steps such as alignment, filtering, and compression before three-dimensional reconstruction and analysis. Similar to other electron microscopy methods, challenges such as charging artifacts and beam damage remain still remain, though advances in staining chemistry, beam optimization, and computational segmentation pipelines continue to improve the accessibility of vEM [7,40,87].

A key strength of vEM is to restore the 3D context often lost in traditional 2D EM, making it possible to visualize the structural relationships across large fields of view. A practical demonstration of different vEM modalities comes from pulmonary tissue studies (Figure 4b) [85]. Using SBF-SEM for the reconstruction of large tissue volumes, the expansive networks formed by alveolar epithelial type 1 cells and their associated capillaries are captured. At higher resolution, FIB-SEM provides superior z-sampling, allowing for nearly complete visualization of alveolar epithelial type 2 cells together with complex surface features such as dense microvilli. Electron tomography pushes this even further, resolving subcellular structures below a few hundred nanometers, including individual lipid membranes and direct connections between lamellar bodies and autophagosomes within AE2 cells. Together, these approaches demonstrate how vEM bridges scales of biological organization, linking organelle-level features with whole-cell and tissue architecture that cannot be captured by conventional 2D EM imaging. In the field of connectomics, the impact of vEM has been transformative, where automated, high-resolution imaging makes it possible to reconstruct and analyze neural microcircuits across large volumes, including the retinal direction-selectivity circuit and extended regions of the neocortex [88,89]. FIB-SEM shows superior z-resolution, which allows for precise quantification of small subcellular structures such as synapses [90]. These applications highlight the versatility of vEM in bridging fine ultrastructural detail with broader tissue-level organization (Table 1).

**Figure 4 nanomaterials-15-01752-f004:**
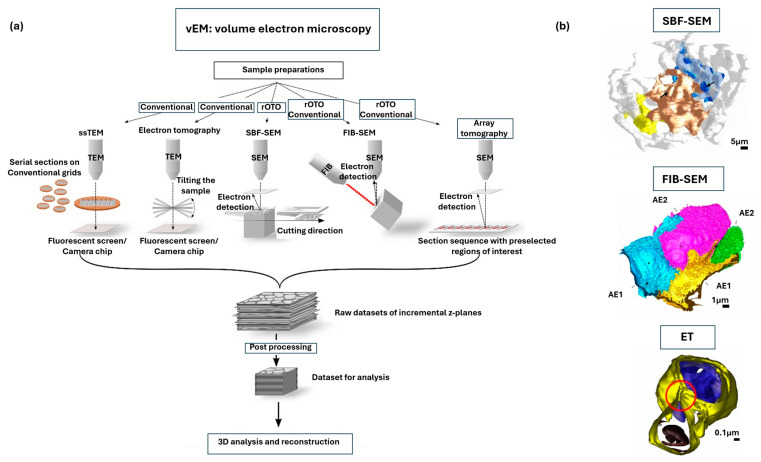
Volume electron microscopy and 3D ultrastructural reconstruction techniques for nanoscale visualization of biological tissues. (**a**) Overview of different volume EM methods, including serial section TEM (ssTEM), electron tomography (ET), serial block-face scanning electron microscopy (SBF-SEM), Focused ion beam scanning electron microscopy (FIB-SEM), and array tomography. Heavy-metal staining protocols, such as reduced Osmium–Thiocarbohydrazide–Osmium (rOTO), can be combined. Adapted with permission from Ref. [85]. Copyright 2020 Springer Nature. (**b**) Different 3D reconstructions examples for volume EM techniques: alveolar epithelial cells and capillary networks (SBF-SEM) (adapted with permission from Ref. [91]; Copyright 2019 American Thoracic Society), whole-cell reconstructions (FIB-SEM) (adapted with permission from [92]; Copyright 2020 MDPI), and organelle connectivity in lung tissue (ET) (adapted with permission from Ref. [93]; Copyright 2015 John Wiley & Sons, Ltd.).

**Table 1 nanomaterials-15-01752-t001:** Comparative overview of major electron microscopy modalities for nanoscale characterizations of biological samples.

Modality	Resolution (Lateral/Axial)	Sample Preparation	Advantages	Limitations	References
SEM	~2–5 nm/Surface only	Fixed, dehydrated; Metal staining/coating	Surface ultrastructure; Compositional contrast	Charging; Dehydration artifacts	[11,90]
TEM	~0.2–2 nm/~1–5 nm	Resin-embed; Ultrathin sections; Heavy-metal staining	Intracellular nanostructures	2D sections; Stain variability; Beam damage	[50,52]
STEM	~1–3 nm/~1–5 nm	Resin-embed; Thin sections; Heavy-metal staining	Z-contrast; Elemental maps	Small field of view; Section artifacts; Stain variability; Beam damage	[57,94]
Cryo-EM	~2–4 Å/3D reconstruction	Vitrified particles on grids; Low dose	Native state; Molecular detail	Grid preparation; Limited throughput	[13,19]
Cryo-ET	~2–4 nm/3D	Vitrified lamellae (~100–200 nm); cryo-FIB	In situ macromolecular context	Lamella thickness limits; Missing wedge effect	[68,95]
vEM	~4–10 nm/~20–50 nm	Heavy-metal staining; Serial sections	Large volumes; Connectomics	Data/time intensive; Registration errors	[11,90]

## 3. Optical Microscopy

Imaging nanoscale features with optical methods has transformed fluorescence microscopy by extending imaging capabilities beyond the diffraction limit while preserving molecular specificity. Collectively, methods such as super-resolution microscopy and expansion microscopy have provided complementary routes to nanoscale resolution, each with distinct principles, strengths, and trade-offs. These techniques not only enable direct visualization of subcellular organizations but also serve as essential counterparts to electron microscopy, offering molecular specificity and compatibility with live specimens. In the following subsections, we outline the major classes of super-resolution and expansion-based approaches, highlighting their operating principles, representative applications, and practical considerations.

### 3.1. Super-Resolution Microscopy

Super-resolution microscopy (SRM) encompasses a variety of advanced optical methods designed to overcome the diffraction limit, a fundamental physical restriction that traditionally limits the optical resolution of fluorescence microscopes to approximately 200–300 nm laterally and 500–800 nm axially [96]. This technological breakthrough has revolutionized the life sciences by enabling the visualization of biological structures at the nanoscale [97,98]. Many biological components, such as subcellular structures like cytoskeletal filaments, synaptic clefts, membrane receptors, and macromolecular complexes, are significantly smaller than this diffraction barrier, often measuring down to tens of nanometers [99]. By surpassing the diffraction limitations, SRM techniques provide unprecedented detail into the molecular architecture and dynamics of cells and tissues. In this section, we focus on three major classes of far-field SRM methods based on single-molecule localization, fluorescence depletion, and structural light illumination.

#### 3.1.1. Single-Molecule Localization Microscopy

Single-molecule localization microscopy (SMLM) is a term that brings together techniques like photoactivated localization microscopy (PALM) and stochastic optical reconstruction microscopy (STORM), relying on the statistical localization of individual molecules [100,101]. The fundamental principle involves distinguishing molecules that occupy the same diffraction-limited volume (Figure 5a) [102]. By actively controlling the fluorescent state of molecules, only a sparse, non-overlapping subset of fluorophores are activated (turned “on”) at any given time. The image produced by a single “on” molecule, while diffraction-limited, has a centroid that can be determined with high precision [103]. By repeatedly cycling fluorophores between a fluorescent state and a non-fluorescent (dark) state across thousands of camera frames, and then mathematically recording the precise coordinates of each localized molecule, a high-resolution, pointillist image is computationally reconstructed [98]. SMLM techniques can achieve very high spatial resolution, typically ranging from 10 to 50 nm [104]. However, SMLM relies on special photoactivatable or photoswitchable fluorophores and specific buffer conditions for robust photoswitching [105]. Long data acquisition time is often required, involving thousands of frames and resulting in acquisition times that can take minutes to generate a single super-resolution image, making imaging of fast dynamic processes challenging [106]. Furthermore, SMLM acquisitions are generally restricted to thin samples or regions close to the coverslip (≤10–20 µm) due to background fluorescence and aberrations.

SMLM has become an indispensable tool in structural cell biology, offering molecular specificity and nanoscale detail that were once beyond the reach of conventional optical microscopy. In one example, STORM resolved synaptic architecture by simultaneously labeling pre- and postsynaptic proteins, with photoswitchable fluorophores, clearly differentiating their nanoscale organization across the synaptic cleft (Figure 5b) [107]. STORM has been applied to systematically compare cytoskeletal components across mesenchymal stem cells, glioblastoma cells, and breast cancer cells, revealing distinct nanoscale differences in actin filament organization and tubulin architecture between cell types (Figure 5c) [108]. STORM has also illuminated cell surface organization by mapping adhesion complexes and revealed protein heterogeneity within the plasma membrane [109]. In neuroscience, SMLM was pivotal in uncovering the quasi-one-dimensional periodic actin–spectrin skeleton in axons, a discovery that reshaped understanding of cytoskeletal organization [110]. Within the nucleus, SMLM has been applied to probe higher-order chromatin dynamics, showing that nucleosomes cluster into compact, coherently moving domains approximately 160 nm in size [111] and visualizing DNA organization within chromatin domains [112]. Applications also extend to virology, where SMLM has resolved the organization of endosomal sorting complex required for transport (ESCRT) subunits at human immunodeficiency virus (HIV) assembly sites [113].

**Figure 5 nanomaterials-15-01752-f005:**
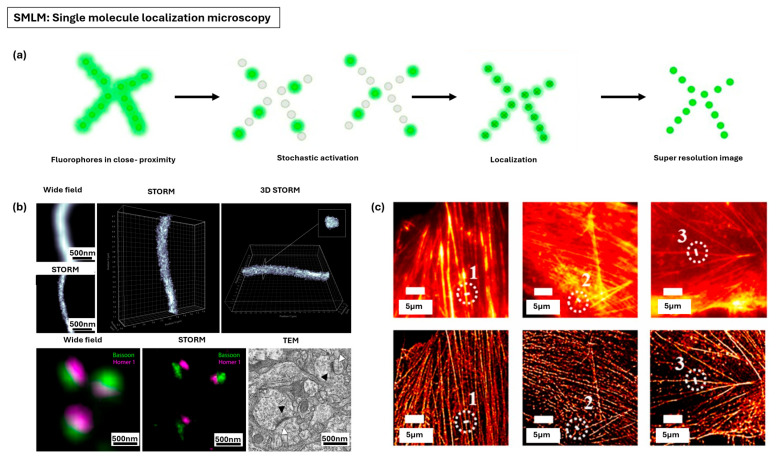
Single-molecule localization microscopy (SMLM) for nanoscale fluorescence imaging. (**a**) Principle of single molecule localization microscopy (SMLM), showing stochastic activation of fluorophores, nanometer precision localization of single emitters, and reconstruction of super-resolved image. Adapted with permission from Ref. [114]. Copyright 2021 Frontiers. (**b**) 3D- and two-color imaging of neuronal structures with stochastic optical reconstruction microscopy (STORM). Adapted with permission from Ref. [107]. Copyright 2020 John Wiley & Sons. (**c**) STORM imaging of cytoskeletal elements in mesenchymal stem cells, glioblastoma cells, and breast cancer cells. Adapted from Ref. [108]. Copyright 2024 Elsevier.

#### 3.1.2. Stimulated Emission Depletion Microscopy

Stimulated emission depletion (STED) microscopy is a super-resolution technique that is based on the platform of confocal laser scanning microscopy (LSM). Its working principle as shown in Figure 6a involves the use of two synchronized laser beams: a conventional Gaussian-shaped excitation beam that initiates fluorescence, and a second, high-intensity depletion beam [99]. The depletion beam is precisely patterned, typically into a doughnut shape with a central intensity minimum, to force excited fluorophores located in the periphery of the focal spot to immediately drop back to the ground state without emitting fluorescence, thus effectively switching them “off” [115]. Only the fluorophores located within the tiny, non-depleted central minimum remain in the excited state long enough to emit detectable spontaneous fluorescence. Since the size of this emission spot can be made arbitrarily small, STED offers diffraction-unlimited resolution [116]. A major advantage of STED is that, as an LSM-based technique, it produces hardware-based images with improved resolution directly, without the mandatory requirement of complex post-processing or reconstruction algorithms. STED is also compatible with a wide range of conventional fluorophores and fluorescent proteins [117]. However, STED requires high depletion laser intensity, which leads to high photobleaching and phototoxicity, and the achievable resolution (routinely 20–70 nm) is ultimately limited by the signal-to-noise ratio due to the constraints on laser power on sample [97].

STED microscopy has proved highly successful and provided spectacular insights, especially in neurocytological applications [117]. One unique advantage of STED is to visualize nanostructures in vivo, and STED has been applied to study the various neuronal structures, including the arrangement of the cytoskeleton in axons, and the plasticity in dendritic spines and synapses [118]. In one example, in vivo STED microscopy has enabled nanoscale imaging of neuronal structures in the living mouse brain and subcellular dendritic branches in layer 1 of the mouse visual cortex were investigated (Figure 6b) [119]. Compared to conventional confocal imaging, STED revealed finer details of actin filament morphology and distribution, achieving a resolution below 80 nm and time-lapse imaging over nearly an hour captured dynamic changes in dendritic spine morphology without detectable phototoxicity. In another example, STED resolved the nanoscale organization of the postsynaptic scaffold protein PSD95 (Figure 6c) and uncovered a striking diversity of postsynaptic density protein 95 (PSD95) arrangements ranging from small round to elongated and complex assemblies [120]. STED has also been utilized in studies focusing on the organization of cell organelles and complexes, such as showing that the pro-apoptotic cell-death mediator Bax forms ring structures on the mitochondrial surface [121]. Due to the inherent optical sectioning capabilities provided by the confocal pinhole, STED is one of the super-resolution modalities most readily to be extended to deep tissue imaging and using an aberration-corrected two-photon excitation-STED, it have enabled imaging of dendritic spines up to 76 μm deep in the brain of a living mouse and 164 μm deep in fixed mouse brain tissue [122].

**Figure 6 nanomaterials-15-01752-f006:**
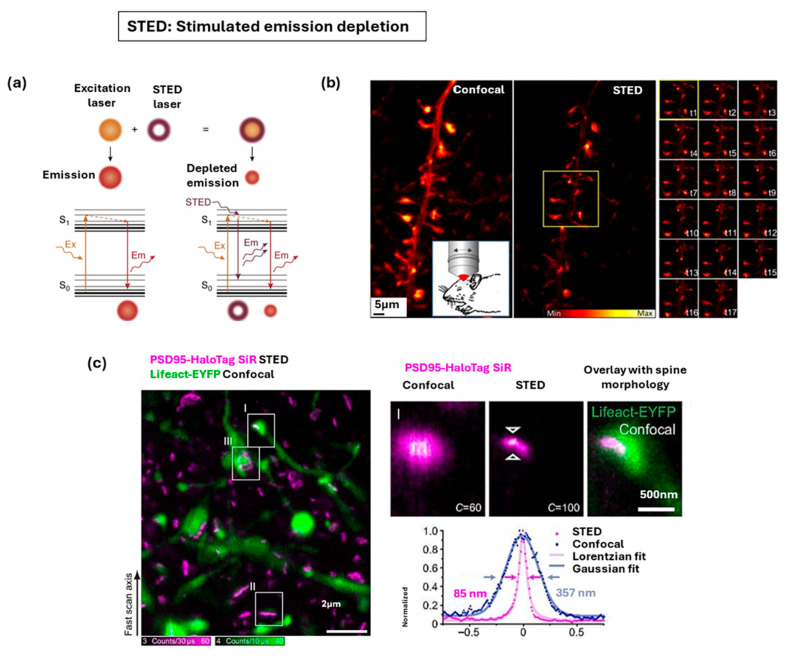
Stimulated emission depletion (STED) microscopy for nanoscale fluorescence imaging. (**a**) Principle of stimulated emission depletion (STED) microscopy. Adapted with permission from Ref. [99]. Copyright 2021 Elsevier. (**b**) In vivo STED imaging of filamentous actin in mouse visual cortex. Adapted from Ref. [119]. Copyright 2017 Springer Nature. (**c**) STED microscopy of postsynaptic density protein 95 (PSD95) in dendritic spines with line profiles confirming improved resolution. Adapted with permission from [120]. Copyright 2018 PNAS.

#### 3.1.3. Structured Illumination Microscopy

Structured illumination microscopy (SIM) illuminates the fluorescent sample with a spatially patterned light (e.g., sinusoidal stripes) that has a frequency close to the diffraction limit (Figure 7a) [99]. This structured pattern creates an interference phenomenon called the Moiré effect, which down-modulates high spatial frequency information, fine structural details usually beyond the microscope’s observable range into the detectable range of the microscope objective [123]. To achieve isotropic resolution, a series of raw images is acquired by rotating the orientation of the illumination pattern and the final super-resolved image is generated through complex computational post-processing, which decodes the Moiré fringes and computationally shifts the high-frequency information back into its correct location in frequency space [124]. SIM generally provides a limited resolution enhancement of only approximately two-fold beyond the diffraction limit, achieving lateral resolution typically around 100–150 nm. Furthermore, achieving high-quality imaging requires collecting a larger number of raw images per focal plane, which reduces the imaging frame rate, and visualize optically thick samples has been challenging, as out-of-focus light can lead to image artifacts [125].

SIM is considered a “gentle” technique suitable for fixed and live-cell imaging and is highly versatile as it works with most conventional fluorescent dyes and proteins [98]. In one example, SIM has been applied to achieve sub-diffraction multicolor imaging of the nuclear periphery, gaining novel insights into the architecture of human centrosomes [126]. In another example, SIM has been applied to visualize three-dimensional structures like liver sinusoidal endothelial cell fenestrations [127]. SIM continues to evolve with technical innovations that address its intrinsic limitations in axial resolution and background suppression. Figure 7b illustrates enhanced resolution and optical sectioning in fixed mouse intestinal organoids, reducing out-of-focus blur [128]. A four-beam SIM has also been developed to achieve near-isotropic super-resolution and applied to live *B. subtilis* membranes, resolving axial features with remarkable precision, detecting fine invaginations (Figure 7c) [129]. These technical advancements have also successfully enabled the capture of three-dimensional super-resolved image stacks of neuronal structures in mouse brain tissue samples for depths exceeding 200 μm, a territory previously beyond the reach of conventional SIM [125].

**Figure 7 nanomaterials-15-01752-f007:**
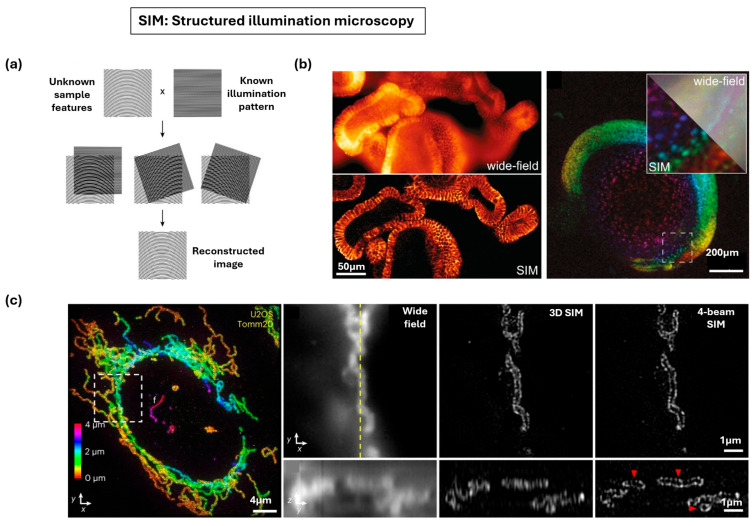
Super-resolution fluorescence microscopy techniques for nanoscale imaging of biological tissues. (**a**) Principle of structured illumination microscopy (SIM). Adapted with permission from Ref. [99]. Copyright 2024 Elsevier. (**b**) SIM shown on fixed intestinal organoids and live zebrafish embryos. Adapted with permission from Ref. [128]. Copyright 2024 Springer Nature. (**c**) Four-beam SIM on cells with mitochondrial labeling demonstrating near-isotropic resolution. Adapted with permission from Ref. [129]. Copyright 2023 Springer Nature.

### 3.2. Expansion Microscopy

Expansion microscopy (ExM) is a technique that achieves super-resolution by physically magnifying biological specimens, enabling nanoscale imaging with conventional, diffraction-limited optical microscopes and the underlying principle is to embed preserved biological material into a dense, swellable hydrogel, such as sodium polyacrylate, that expands isotropically in three dimensions [130]. As illustrated in Figure 8a, the process begins with sample fixation and permeabilization, and target molecules or fluorescent labels are then covalently anchored to the polymer network so that their relative positions remain faithfully preserved during expansion. Once the hydrogel is polymerized, the tissue becomes embedded within a cross-linked, water-absorbent scaffold. To ensure uniform swelling, the native crosslinks of the specimen are homogenized through enzymatic digestion or chemical denaturation, like proteinase K, which dissolves rigid structures while leaving the anchored labels intact. Immersion in water or a low-osmolarity buffer subsequently drives the hydrogel to swell, expanding the specimen physically by about 4–5× in all directions [131]. This magnification effectively improves the diffraction limit from ~300 nm to ~70 nm, allowing nanoscale details to be resolved with standard optical microscopy. ExM can be directly adapted to intact multicellular systems, which allows whole-organism samples such as zebrafish larvae, Drosophila wing discs, and mouse embryos to be expanded ~4× isotropically (Figure 8b) [132,133]. In addition to this resolution enhancement, ExM stands out for its accessibility and low cost, relying only on common laboratory reagents, and the expanded samples are highly transparent, enabling deep 3D imaging of thick tissues and large volumes [29,134]. Building upon the physical expansion principle, iterative ultrastructure expansion microscopy (iU-ExM) allows repeated rounds of gel embedding and expansion, pushing resolution further, further extending the technique by preserving near-native ultrastructure and enabling molecular-scale resolution, reaching a precision comparable to SRM (Figure 8c) [135,136,137]. In an example of visualizing human nuclear pore complex, conventional fluorescence shows these complexes as unresolved spots, ultrastructure expansion microscopy (U-ExM) shows ring-like arrangements, and iU-ExM ultimately resolves individual NPC subunits using a standard widefield system (Figure 8c) [136].

ExM has applications across a wide spectrum of biological fields, ranging from clinical diagnostics to fundamental biological imaging. In medicine, ExM has enabled nanoscale analysis of formalin-fixed, paraffin-embedded tissues, supporting the diagnosis of kidney minimal-change disease and to differentiate early breast neoplastic lesions, applications that were previously limited to electron microscopy [140]. Beyond pathology, ExM has been successfully adapted to diverse imaging contexts, including the visualization of bacterial ultrastructure and viral infection processes within host cells, imaging of plant tissues such as roots and cell walls [141,142]. In neuroscience, light-microscopy-based connectomics has combined engineered iterative hydrogel expansion (~16×) with deep-learning–based segmentation, enabling dense reconstruction of mouse cortical circuits at synaptic resolution (Figure 8d) [138]. This approach not only captures neuronal architecture such as axons, dendrites, and spines but also integrates molecular predictions of synaptic proteins like bassoon and SHANK2, bridging structural and molecular scales. To validate expansion fidelity, complementary tissue-specific protocols have been developed, as demonstrated in mosquito salivary glands by comparing pre- and post-expansion states stained for DNA and lipids (Figure 8e) [139]. Collectively, these advances show how ExM-derived approaches are reshaping nanoscale imaging, making it possible to study ultrastructure and molecular composition across intact tissues and even at the connectome level. At the same time, ExM is being integrated with advanced optical methods discussed earlier including SMLM, STED, SIM, and LSM that push the achievable resolution even further, sometimes revealing ultrastructural features such as centriolar chirality that were once only accessible with electron microscopy (Table 2) [143].

## 4. Mechanical and Chemical Nanoscale Imaging of Biological Samples

Beyond electron and optical approaches, several other techniques have been developed to probe the nanoscale composition of biological tissues. Atomic Force Microscopy (AFM) provides topographical and mechanical measurements with nanometer precision, enabling quantification of cell stiffness, viscoelasticity, and extracellular matrix mechanics under both physiological and pathological conditions [149,150,151,152,153,154,155]. Nanoscale secondary ion mass spectrometry (nanoSIMS) extends analysis into the chemical domain, offering isotopic and elemental mapping with sub-50 nm spatial resolution [156,157,158,159,160,161,162]. Tip-enhanced Raman spectroscopy (TERS) adds a complementary vibrational dimension, enabling label-free chemical imaging of biomolecules such as proteins, lipids, and nucleic acids with spatial resolution far beyond the diffraction limit of conventional Raman microscopy. These other imaging modalities provide complementary mechanical and chemical insights that enrich nanoscale characterization and analytical toolkit beyond electron and optical imaging modalities discussed in Section 2 and Section 3.

### 4.1. Atomic Force Microscopy

Atomic force microscopy (AFM) is a scanning probe technique that provides three-dimensional visualization of biological samples at sub-nanometer resolution, while also enabling quantitative measurements of their mechanical properties under physiologically relevant conditions [163,164,165]. As the sharp probe mounted on a flexible cantilever scans across the sample, these local forces between the probe and the sample surface cause the cantilever to bend, and the resulting deflections are detected by a laser beam reflected onto a position-sensitive photodetector (Figure 9a) [166,167]. This optical lever system translates cantilever movements into measurable optical signals, which, combined with piezoelectric scanners for precise xyz positioning, reconstruct high-resolution maps of surface topography. Beyond topography imaging, AFM also acts as a force sensor and nano indenter, generating force–distance curves as the tip approaches and retracts from the sample. By fitting these data to contact mechanics models, parameters such as Young’s modulus can be extracted, enabling quantitative mechanical profiling [168,169], and machine learning makes data interpretation more efficient [170]. Importantly, AFM does not require chemical staining or fixation, and it can operate in liquid environments, preserving the native state of living or hydrated tissues [171]. Modern AFM further extends these capabilities by developing different modalities, such as PeakForce tapping mode and ringing mode, which measures adhesion, dissipation energy, height, restored adhesion, and disconnection energy loss [172]. As shown in Figure 9b, when applied to biological tissues like human skin flakes and melanoma cells, ringing mode gives nanoscale details of thin surface layers with greater clarity and reduced artifacts [172].

AFM have linked nanoscale morphology and mechanics to biological function [151]. In static biological samples, AFM has been used to quantify stiffness, viscosity, adhesion, and mechanical profiling [152,173]. For dynamic processes, high-speed AFM can capture moving trajectories of proteins and their morphology fluctuations in liquid environment [174,175]. Due to the unique flexibility in probe configuration, AFM can be combined with other optical modalities to expand functional and chemical analysis [176,177]. AFM has also been applied to various clinical applications for diagnosis. For example, AFM has been applied to cancer tissue and detected nanomechanical tumor signatures [178,179]. AFM has also been applied to investigate age-related changes in biological tissue, such as the change in collagen fibril periodicity in connective and mineralized tissues during aging [20,180] and amyloid-β aggregation in Alzheimer’s disease [181,182].

### 4.2. Nanoscale Secondary Ion Mass Spectrometry

Nanoscale secondary ion mass spectrometry (nanoSIMS) is a powerful chemical imaging technique that provides quantitative, high-resolution maps of elemental and isotopic distributions within biological samples. Unlike optical and electron microscopy, which primarily resolves morphology, nanoSIMS directly visualizes the molecular and isotopic composition of cells and tissues with subcellular precision [183,184]. Figure 10a provides a schematic overview of the nanoSIMS instrument, which uses a finely focused primary ion beam (commonly Cs^+^ or O^−^) to sputter atoms and molecules from the sample surface in ultra-high vacuum [185]. These ejected secondary ions are collected and separated in a magnetic sector mass spectrometer with multiple detectors, enabling simultaneous measurement of up to 5–7 isotopes at lateral resolutions approaching 50 nm [186], making nanoSIMS particularly suitable for stable isotope tracing, where isotope-labeled metabolites (e.g., ^15^N, ^13^C, ^2^H) are incorporated into cells and tissues [187]. A key strength of nanoSIMS is its correlative potential: When integrated with electron or fluorescence microscopy, it overlays ultrastructural information with molecular identity, bridging form and function in a way few other modalities can achieve [188]. For example, coupling nanoSIMS with TEM or STED optical microscopy can provide multi-modal views where ultrastructural detail is directly linked to isotope distributions [189]. Improved detectors also allow simultaneous multi-isotope imaging with greater sensitivity, together with computational advancement that enables better quantification of isotope ratios and faster image analysis pipelines [190].

Biological applications of nanoSIMS have expanded rapidly, especially in studying the metabolic research. NanoSIMS can map lipid metabolism at nanometer resolution, such as tracing isotope-labeled fatty acids into cytosolic lipid droplets of adipocytes and cardiomyocytes [191]. In one example, nanoSIMS is used to trace the distribution of stable isotope labeled lipids in brown adipose tissue and heart following different administration routes (Figure 10b) [185]. For mice gavage with ^13^C-labeled fatty acids, nanoSIMS images combining ^12^C^14^N^−^ (for morphology) with ^13^C/^12^C ratio maps revealed enrichment of ^13^C within cytosolic lipid droplets, nearly double the natural abundance. These results emphasize the strength of nanoSIMS for visualizing lipid trafficking and metabolism in situ, combining stable isotope labeling with nanoscale chemical imaging. In microbiology, nanoSIMS has been used to quantify single-cell carbon and nitrogen fixation, enabling studies of nutrient cycling and symbiotic exchange that were previously inaccessible [192]. In neurobiology and disease modeling, NanoSIMS has revealed pathological changes in organelle-specific metabolism, like α-synuclein-induced defects in mitochondria and Golgi in Parkinson’s disease tissues [193].

### 4.3. Tip-Enhanced Raman Spectroscopy

Tip-enhanced Raman spectroscopy (TERS) is an advanced near-field vibrational imaging technique that can be used for label-free, high-resolution chemical mapping of biological samples by overcoming the diffraction-limited spatial resolution of conventional Raman microscopy [194,195,196,197,198,199,200,201]. While far-field Raman can identify biomolecular signatures, its spatial resolution is typically limited to several hundred nanometers, restricting its ability to probe the nanoscale heterogeneity inherent to cell membranes, protein complexes, and subcellular structures. Figure 11a illustrates the TERS principle, in which an excitation laser illuminates a sharp metallic AFM tip brought into nanometer proximity with the sample surface [202]. The confined optical near-field at the tip apex enhances Raman scattering from molecules directly beneath the tip, providing chemical sensitivity at the 10–20 nm scale which is an order-of-magnitude improvement over confocal Raman imaging. The optical performance of TERS strongly depends on illumination geometry, as shown in Figure 11b [203]. Bottom-illumination offers high enhancement efficiency but is restricted to transparent substrates, whereas side- and top-illumination geometries accommodate opaque biological tissues and require long-working-distance objectives for stable excitation and collection.

Biological applications of TERS have expanded significantly as instrumentation and probe stability have improved. Figure 11c shows a representative example in which correlative AFM, confocal Raman, and TERS imaging are used to map the biochemical organization of a BxPC-3 human pancreatic cancer cell membrane [204]. Confocal Raman images constructed from protein marker bands such as phenylalanine (1001 cm^−1^) and histidine (1532 cm^−1^) exhibit little spatial contrast due to diffraction-limited sampling. In contrast, TERS images can effectively identify the pronounced nanoscale variations, resolving chemically distinct domains separated by only tens of nanometers. Extracted spectra from high- and low-intensity TERS pixels further confirm localized enrichment of specific amino-acid signatures, underscoring the ability of TERS to detect molecular heterogeneity at the level of individual membrane components. These capabilities make TERS particularly valuable for investigating cell-membrane organization, protein aggregation, amyloid formation, bacterial envelope chemistry, and extracellular matrix remodeling. Continued advances in probe design, drift-correction, multi-modal AFM-TERS integration, and machine-learning-assisted spectral analysis are expected to further expand the role of TERS in nanoscale tissue characterization, enabling researchers to link biochemical identity with structural context in complex biological systems. Furthermore, with the advancement in correlative imaging techniques, Raman imaging, fluorescence microscopy, and electron microscopy can complement each other and clarify morphological, structural, and chemical information of materials at the nanoscale (Table 3) [205,206].

## 5. Artificial Intelligence in Nanoscale Tissue Characterization

Artificial intelligence (AI) is redefining nanoscale tissue characterization by tackling one of the field’s central challenges: the analysis of massive, information-rich three-dimensional imaging datasets. Modern imaging modalities such as electron microscopy, super-resolution microscopy, and expansion microscopy routinely generate terabyte-scale volumes, capturing biological detail down to the molecular level. Extracting meaningful insights from such data is beyond the capabilities of conventional image processing and manual annotation. AI-driven methods, particularly those based on deep learning, now provide the tools to segment complex tissue architectures, trace subcellular structures, and reconstruct three-dimensional volumes with unprecedented accuracy and speed. These approaches not only reduce human bias and workload but also enable new biological discoveries by linking ultrastructural morphology with functional context. Importantly, AI is shifting nanoscale imaging from primarily descriptive visualization toward quantitative, high-throughput analysis.

### 5.1. Segmentation

Segmentation refers to the process of accurately delineating and quantifying individual cells, subcellular structures, or even organelles within complex volumetric datasets [210]. Imaging technologies alone, whether confocal, light-sheet, or electron microscopy, primarily provide raw visual information, and these images remain descriptive snapshots rather than quantitative resources. In developmental biology and morphogenesis research, segmentation enables the extraction of precise information about cell shape, size, orientation, and spatial relationships across tissues and time, allowing researchers to transform imaging data into measurable biological insights. In neuroscience, segmentation is crucial for reconstructing neuronal circuits and quantifying synaptic connections at the nanoscale [211]. The challenge is scale and complexity since modern imaging platforms routinely produce terabyte-scale datasets of developing tissues or entire organs, capturing millions of densely packed cells. Converting this raw data into traceable, quantifiable objects has become a critical bottleneck in analysis, since manual annotation is prohibitively time-consuming and subjective. Early computational efforts used classical computer vision tools such as edge-detection algorithms [212] and combination of simple image filters with machine learning [213]. While these methods offered partial solutions, they often failed in dense or heterogeneous tissues where cell boundaries are ambiguous or overlapping. To overcome these limitations, more advanced strategies emerged from connectomics in electron microscopy, particularly graph-based approaches like affinity graph learning [214], multicuts [215], and region adjacency graph methods [216]. These methods reframed segmentation as a global optimization problem, enabling more robust and consistent boundary detection even in challenging biological contexts.

Deep learning has fundamentally advanced segmentation by introducing highly accurate, automated pipelines that approach human-level performance. The U-Net architecture [217], and later its volumetric extension, the 3D U-Net [218], defined the standard for biomedical image segmentation. The segmentation workflow typically follows a two-step pipeline, as shown in Figure 12a [210]. In the first stage, a convolutional neural network like 3D U-Net predicts cell boundaries, producing probability maps that capture membranes or walls within dense tissue volumes. These boundary predictions are then refined through graph partitioning algorithms such as watershed [219], multicuts [215], or Mutex watershed [220], which group adjacent voxels into discrete cellular objects. The result is a complete segmentation map in which each cell or subcellular unit is uniquely labeled. Figure 12b highlights the effectiveness of this approach, showing dense 3D reconstructions of complex nanoscale structures such as axons, neurites, or dendrites [138]. By enabling clear separation of individual components within crowded environments, segmentation pipelines transform raw volumetric imaging into quantitative datasets suitable for morphodynamical analysis and nanoscale connectivity mapping. In connectomics, specialized networks such as flood-filling networks [221] have enabled the reconstruction of complex neuronal circuits with remarkable precision and these advances have been adapted into pipelines for volumetric light microscopy data at a level comparable to electron microscopy [138].

### 5.2. Image Reconstruction

Image reconstruction in AI-enhanced nanoscale tissue characterization is a computational strategy that compensates for deliberate compromises made during image acquisition. The principle is to sacrifice certain parameters, such as the point spread function (PSF), SNR, sampling density, or pixel resolution, to increase imaging speed, reduce data burden, or minimize phototoxicity in delicate biological specimens like living tissues. These trade-offs are then computationally corrected using deep learning models trained on large datasets of high-quality images, enabling the recovery of fine structural detail. By doing so, AI-driven reconstruction restores critical imaging metrics, such as spatial and temporal resolution, field of view, depth of field, and space–bandwidth product, that would otherwise be lost. This approach directly addresses the major challenges of nanoscale imaging, where prolonged exposure and large data volumes are often unavoidable. Historically, surpassing the diffraction limit required complex optical methods like STED microscopy [115,222] or SMLM approaches [100,101], which imposed high light dosage and were limited in live-cell contexts. AI-based image reconstruction thus represents a paradigm shift, allowing researchers to achieve nanoscale resolution with greater efficiency and gentleness.

AI-based image reconstruction can dramatically reduce the data burden of volumetric imaging by undersampling the axial dimension as shown in Figure 13a [223]. Instead of acquiring dense stacks of images, only a sparse subset of planes is collected, significantly decreasing acquisition time and light exposure. A recurrent convolutional neural network like Recurrent-MZthen infers the missing information, propagating these sparse 2D inputs into a complete 3D volume, thereby extending the depth of field and reducing the number of scans required by more than an order of magnitude. Figure 13b demonstrates how deep learning further compensates for low SNR inputs, a trade-off often made to minimize phototoxicity in sensitive live-cell imaging [223]. In this case, low-exposure STED images are restored to high fidelity using networks such as U-Net or RCAN, which are trained on paired low-SNR and ground-truth datasets. This approach reduces photon load by nearly 90% while increasing acquisition speed several-fold, producing reconstructions that faithfully approximate high-quality inputs. The combination of these strategies is shown in Figure 13c, where the integration of undersampling, restoration, and segmentation yields dense 3D reconstructions of complex nervous tissue [211]. Distinct nanoscale components including unmyelinated axons, spiny dendrites, myelinated axons, and glial fragments are clearly resolved, enabling quantitative structural analysis such as connectivity mapping and morphodynamical studies.

Innovations in AI reconstruction have spanned across diverse imaging modalities. Deep learning models based on convolutional neural networks now enable effective SNR enhancement, restoring high-fidelity images from low-exposure inputs and reducing photon load in sensitive applications like live-cell STED microscopy [224] and nanoparticle imaging with EM [225]. Other methods refine distorted PSFs to achieve rapid autofocusing [226] and deblurring [227], which recovers sharp images from motion-blurred scans to facilitate ultra-fast acquisition. Sparse-data reconstruction frameworks, such as recurrent neural networks, can digitally reconstruct full 3D sample volumes from only a limited number of 2D images, drastically cutting acquisition times [228]. Physics-informed models like the enhanced Fourier imager network further integrate super-resolution and autofocusing for holographic imaging, pushing the boundaries of label-free volumetric microscopy [229]. Together, these advances demonstrate that AI-based reconstruction not only alleviates hardware and acquisition constraints but also enables new classes of experiments, such as long-term live imaging and nanoscale mapping of large volumes.

## 6. Conclusions

Nanoscale characterization of biological tissues has progressed from proof-of-concept demonstrations to standard protocols for modern high-resolution biological imaging. This review has discussed four major technical areas: Electron microscopy, which provides high-resolution structural mapping; optical nanoscopy, which delivers molecularly specific views below the diffraction limit; other experimental modalities, such as atomic force microscopy, tip-enhanced Raman spectroscopy and nanoscale secondary ion mass spectrometry, which extend imaging to mechanical and chemical dimensions; and AI tools, which enable transformation of raw datasets into quantitative biological information. Rather than relying on a single imaging modality, emerging studies integrate multiple approaches into correlative pipelines that link structural, chemical, and functional information across scales. As imaging modalities and computational pipelines continue to merge, nanoscale tissue characterization is moving beyond descriptive mapping toward mechanistic insight into development, pathology, and therapeutic response, thereby transforming how biological systems are visualized and understood across molecular, cellular, tissue, and organ scale.

## Figures and Tables

**Figure 1 nanomaterials-15-01752-f001:**
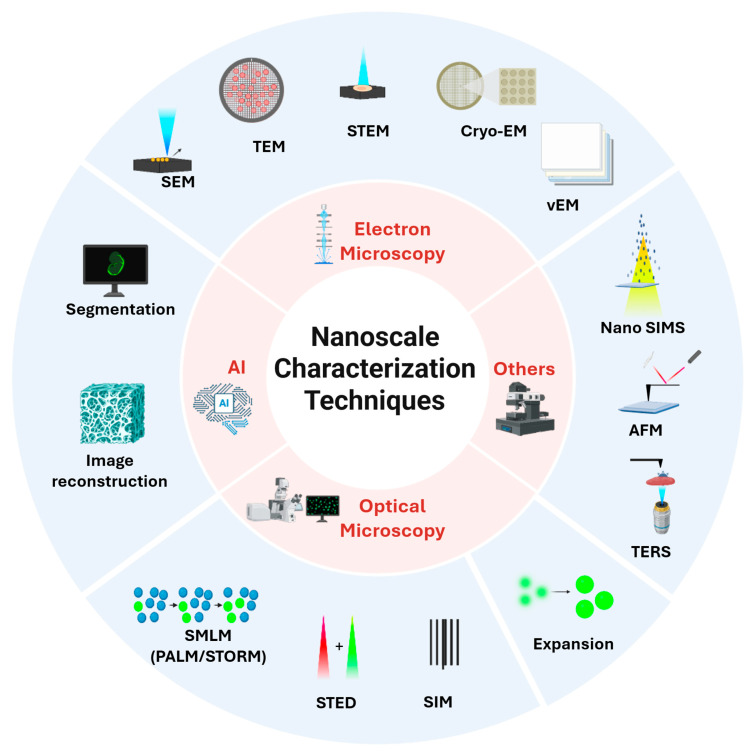
Nanoscopic characterization techniques for biological tissues. The review is structured into four sections: Electron microscopy, optical microscopy, mechanical and chemical mapping tools, and AI-assisted data analysis.

**Figure 8 nanomaterials-15-01752-f008:**
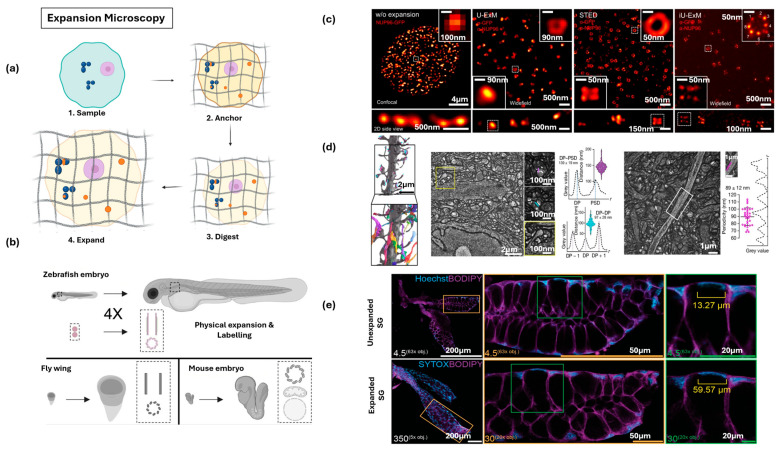
Expansion microscopy approaches for nanoscale imaging of biological tissues. (**a**) Conceptual workflow of expansion microscopy (ExM), highlighting anchoring of biomolecules, enzymatic digestion, and physical expansion. (**b**) Schematic illustrating physical expansion and labeling applied to zebrafish embryos (4×), fly wings, and mouse embryos, enabling nanoscale imaging of intact tissues. Adapted with permission from Ref. [132]. Copyright 2023 Springer Nature. (**c**) Iterative ultrastructure expansion microscopy (iU-ExM) of human nuclear pore complexes, shown alongside confocal, ultrastructure expansion microscopy (U-ExM), and stimulated emission depletion microscopy (STED), highlighting 8-fold symmetry in top and side views. Adapted with permission from Ref. [136]. Copyright 2023 Springer Nature. (**d**) Dense connectomic reconstruction of mouse somatosensory cortex after ~16× expansion, showing large-scale cell mapping and nanoscale views of dendrites, spines, and synapses. Adapted with permission from Ref. [138]. Copyright 2025 Springer Nature. (**e**) ExM applied to Drosophila salivary glands, comparing unexpanded and expanded samples stained for DNA and lipids, demonstrating clear enlargement of the same tissue structures. Adapted with permission from Ref. [139]. Copyright 2024 Springer Nature.

**Figure 9 nanomaterials-15-01752-f009:**
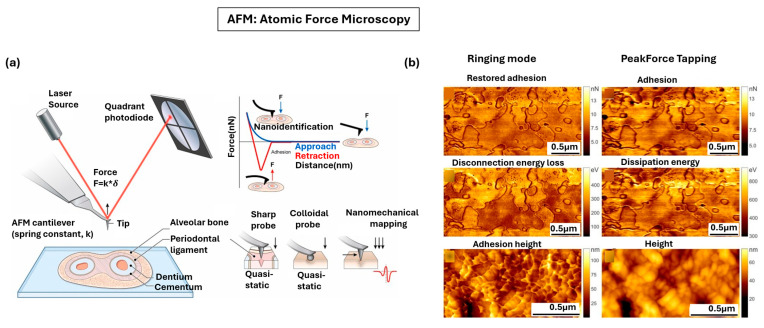
AFM for nanoscale mapping of biological tissues. (**a**) Schematic of AFM showing laser reflection from a cantilever tip and detection of deflections by a photodiode, together with force–distance curves illustrate indentation and adhesion. Adapted from Ref. [166]. Copyright 2023 Elsevier. (**b**) AFM maps of human skin flakes and melanoma cells with two modes: Ringing mode highlights restored adhesion, disconnection energy loss, and adhesion height; PeakForce tapping providing adhesion, dissipation energy, and height maps. Adapted with permission from Ref. [172]. Copyright 2017 Springer Nature.

**Figure 10 nanomaterials-15-01752-f010:**
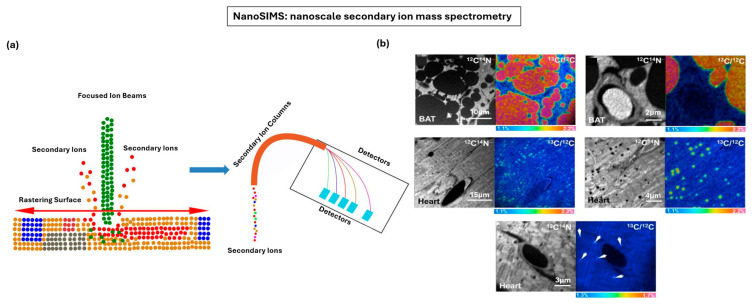
NanoSIMS for nanoscale chemical mapping of biological tissues. (**a**) Schematic of the nanoSIMS instrument showing a focused primary ion beam (Cs^+^ or O^−^) bombarding the sample surface to release secondary ions, which are collected and analyzed by a mass spectrometer for high-sensitivity chemical mapping. (**b**) NanoSIMS imaging of ^13^C-labeled lipids in brown adipose tissue and heart, from mice fed ^13^C fatty acids or injected with ^13^C triglyceride-rich lipoproteins, showing ^13^C/^12^C ratio images that highlight lipid droplet enrichment and metabolic incorporation of labeled lipids. (**a**,**b**) Adapted with permission from Ref. [185]. Copyright 2014 Elsevier.

**Figure 11 nanomaterials-15-01752-f011:**
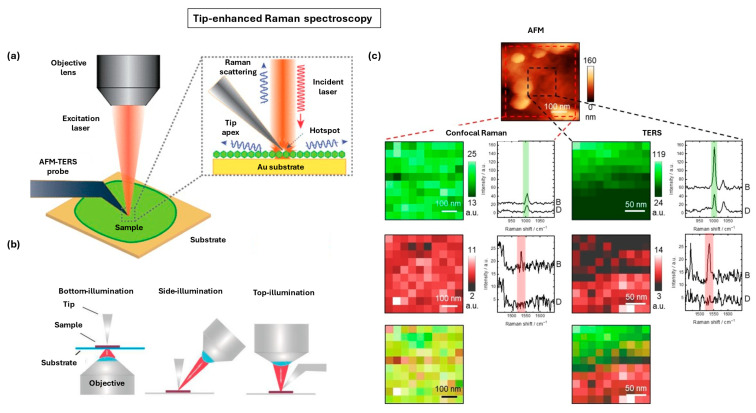
Tip-enhanced Raman spectroscopy (TERS) for nanoscale characterization of biological tissues. (**a**) Schematic representation of TERS showing an excitation laser illuminating an AFM tip positioned close to the sample surface. Adapted with permission from Ref. [202]. Copyright 2021 American Chemical Society. (**b**) Illustration of commonly used illumination geometries in TERS. Adapted with permission from Ref. [203]. Copyright 2018 MDPI. (**c**) Correlative AFM, confocal Raman, and TERS imaging of the cell membrane. AFM topography (top) identifies nanoscale surface features, while confocal Raman maps show weak, spatially uniform signals and corresponding TERS maps reveal strong near-field enhancement and nanoscale chemical contrast. Adapted with permission from Ref. [204]. Copyright 2022 Wiley-VCH GmbH.

**Figure 12 nanomaterials-15-01752-f012:**
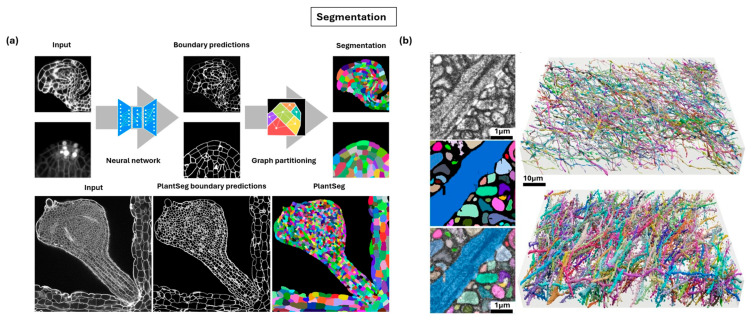
AI-driven volumetric segmentation for nanoscale visualization of biological tissues. (**a**) AI-driven volumetric segmentation workflow: using neural networks to generate boundary predictions and refined using graph partitioning steps to achieve precise segmentation of cellular and subcellular structures. Adapted with permission from Ref. [210]. Copyright 2020 eLife. (**b**) AI-based volumetric segmentation applied to nervous tissue, transforming raw data into distinct, traceable structures such as axons and neurites for quantitative analysis. Adapted with permission from Ref. [138] Copyright 2025 Springer Nature.

**Figure 13 nanomaterials-15-01752-f013:**
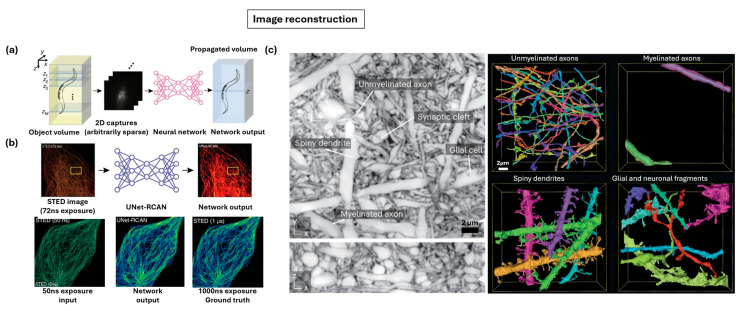
AI-driven volumetric segmentation and image reconstruction for nanoscale visualization of biological tissues. (**a**) AI-based Image reconstruction schematic, transforming sparse 2D axial inputs into a complete 3D volume to speed up acquisition while retaining nanoscale detail. (**b**) AI-based image restoration for volumetric reconstruction, converting low signal inputs into high-fidelity outputs that improve resolution while reducing photon load. (**a**,**b**) Adapted with permission from [223]. Copyright 2024 Springer Nature. (**c**) High-fidelity nanoscale visualization of dense neural tissue after AI-based reconstruction, segmentation, distinguishing unmyelinated and myelinated axons, spiny dendrites, and glial fragments. Adapted with permission from Ref. [211]. Copyright 2023 Springer Nature.

**Table 2 nanomaterials-15-01752-t002:** Summary of optical microscopy and physical expansion techniques for nanoscale imaging of biological samples.

Method	Effective Resolution (Lateral/Axial)	Labels/Preparation	Advantages	Limitations	Key References
SMLM	~10–20 nm/~30–60 nm	Photoswitchable fluorophores; Buffer-dependent; Long imaging series	Highest resolution; Mapping single molecules	Slow acquisition; Drift-prone; Thin samples	[100,101,103,144]
STED	~30–70 nm/~50–150 nm	Conventional dyes and fluorescent proteins; High-intensity depletion laser	Fast imaging; Live-cell and in vivo imaging	Photobleaching; Phototoxicity; Limited multiplexing	[118,120,145,146]
SIM	~100–120 nm/~250–300 nm; ~160–180 nm axial (4-beam SIM)	Conventional dyes and fluorescent proteins; Patterned illumination	Gentle imaging; Large fields of view	Limited resolution gain (~2×); Reconstruction artifacts	[128,129]
ExM	~60–70 nm (4× expansion)	Label features of interests; Anchor to swellable gel, digest, and expand	Large volumes; Cheap	Distortion and anisotropy; Complex protocols	[130,147,148]

**Table 3 nanomaterials-15-01752-t003:** Summary of AFM and nanoSIMS for nanoscale tissue characterization.

Technique	Resolution/Depth	Measurement	Advantages	Limitations	Key References
AFM	~1–10 nm lateral; Sub-nm depth	Surface topography; Nanomechanical properties	Nanomechanical mapping; Hydrated, soft biological samples	Surface-only; Tip convolution artifacts; Sensitivity to environmental drift	[166,207]
nanoSIMS	~50–100 nm lateral; Depth by ion sputtering	Elemental and isotopic mapping	Isotope tracing; Metabolic activity	Charging in non-conductive samples; Destructive imaging	[159,185,208]
TERS	~10–20 nm lateral; Surface-sensitive	Molecular vibrational fingerprints	Label-free chemical specificity; Nanoscale correlative mapping	Requires metallic tip; Sensitive to drift and alignment; limited field-of-view	[204,209]

## Data Availability

No new data were created or analyzed in this study. Data sharing is not applicable to this article.

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
