# Peer review of "Nanoscale Imaging of Biological Tissues: Techniques, Challenges and Emerging Frontiers"

_nanomaterials, 2025, doi:10.3390/nano15231752_

Round 1
Reviewer 1 Report
Comments and Suggestions for Authors
See attached Word document.

Author Response
Comment 1: The paper attempts to review several imaging methods used in biology and the authors group them into three sections: electron microscopy, optical nanoscopy, and mechanical and chemical methods, where the authors discuss atomic force microscopy and nanoscale secondary ion mass spectrometry. The last section discusses machine learning (AI) methods used in image segmentation and reconstruction.
Response: Thank you for summarizing the structure and scope of our manuscript. We are glad that the organization into electron microscopy, optical nanoscopy, mechanical/chemical methods, and AI-based image analysis was clear.
Comment 2: In the author’s words the “review is intended for both experimentalists and computational scientists who require a practical, side-by-side understanding of what each method offers, how to prepare and handle tissue samples, which pitfalls to avoid, and how modern AI pipelines transform raw image stacks into quantitative insights”.
Response: Thank you for highlighting this point. We appreciate the reviewer’s attention to the intended scope and audience of the review.
Comment 3: I am afraid the review falls short of the promise, as each section scratches the proverbial surface and does not go into depth and only “showcases” the abovementioned methods. The figures are packed to the brim and are hard to follow (apart from Figure 1). One can contrast the packed figures with quite a useful Table 1 in Section 2 that summarizing key features and limitations of various electron microscopy methods. Perhaps the most useful part of the review is an extensive bibliographical list with over 200 references.
Response: Thank you for this constructive feedback. We appreciate the reviewer’s concern regarding depth and figure readability. We have changed the figures in revised manuscript.
Our modification to the manuscript: We have expanded the content in each section and significantly improved figure clarity by splitting the original composite figures into multiple focused figures. Specifically, the original Figure 2 (electron microscopy) has been divided into three independent figures, Figure 3 (optical nanoscopy) into three figures, Figure 5 (AFM and nanoSIMS) into two figures, and Figure 6 into two figures. We have also added a new figure illustrating tip-enhanced Raman spectroscopy (TERS). In addition, we incorporated a dedicated discussion paragraph on correlative light and electron microscopy (CLEM) and added a new subsection on environmental SEM (ESEM). These revisions collectively enhance depth, readability, and conceptual clarity across the manuscript.
Comment 4: Such a review, by its nature, shows the author’s preferences and the authors faithfully acknowledge it that they “do not attempt to provide an exhaustive catalogue of all nanoscopy modalities”. In the reviewer’s view, the biggest omission is the absence of scattering methods, which are now prevalent in structural biology, see e.g. a review by Trewhella, Structure (2021), Volume 30, , 15 – 23 (doi: 10.1016/j.str.2021.09.008).
Response: Thank you for raising this point and for highlighting the importance of small angle scattering techniques in structural biology. We fully agree that SAXS and SANS provide powerful insights into biomolecular assemblies and have had significant impact, as outlined in Trewhella (2021). In this review, we focus on real-space direct imaging methods and scattering techniques, such as small angle x-ray and neutron scattering, which probe solution-averaged structures of biomolecules and complexes, are not included.
Our modification to the manuscript: The aim of our paper is to discuss several of the most widely used and practically important nanoscale characterization techniques for biological tissues. While we do include certain scattering-based approaches such as SEM, our focus is specifically on methods that provide direct, spatially resolved nanoscale imaging of intact tissues. For this reason, we did not cover small-angle X-ray or neutron scattering methods, which yield ensemble-averaged structural information in solutions rather than real-space images.
To make this scope clear to readers, we added a brief statement in the Introduction “In this review, we focus on real-space direct imaging methods and scattering techniques, such as small angle x-ray and neutron scattering, which probe solution-averaged structures of biomolecules and complexes, are not included.”
At the same time, we have also changed the title of our manuscript to “Nanoscale imaging of biological tissues: Techniques, challenges and emerging frontiers” to narrow the scope of techniques discussed in our review.
Comment 5: One also wonders, if yet another review is needed as there are already a number of similar reviews, see for example an excellent review article on electron microscopy by Kourkoutis, Plitzko, and Baumeister in Annual Review of Materials Research (2012), Vol. 42, 33-58 (doi: 10.1146/annurevmatsci-070511-155004).
Response: Thank you for this comment and for pointing us to the excellent review by Kourkoutis, Plitzko, and Baumeister. We appreciate the concern regarding the need for an additional review in this area. Our intention with this work is not to duplicate existing single-modality reviews, but rather to bring together multiple major classes of nanoscale characterization techniques, electron microscopy, optical nanoscopy, mechanical and chemical probes, and modern AI-based analysis methods, within a single, tissue-centered framework. While high-quality reviews exist for individual modalities, we found that a resource integrating EM, super-resolution optical methods, AFM, nanoSIMS, TERS, and computational reconstruction pipelines in one place is currently lacking. We have clarified this distinctive integrative scope in the Introduction and Conclusion, emphasizing that the manuscript aims to serve readers who seek a consolidated overview of complementary nanoscale imaging methods specifically in the context of biological tissues.
Comment 6: In order to improve readability and general usefulness, I would recommend splitting each figure (except Figure 1) into several more focused figures and placing them withing a relevant subsection and with much shorter captions.
Response: Thank you for this helpful suggestion. We agree that separating the multi-panel figures would significantly improve clarity and readability.
Our modification to the manuscript: In the revised version, we have split all composite figures into multiple focused figures, each positioned within its relevant subsection and accompanied by much shorter, clearer captions. Specifically, the original Figure 2 (electron microscopy) was divided into three independent figures, Figure 3 (optical nanoscopy) into three figures, Figure 5 (AFM and nanoSIMS) into two figures, and Figure 6 into two figures. We also added an additional figure illustrating tip-based Raman/TERS for improved clarity. These changes substantially enhance readability and align each figure more closely with the corresponding methodological discussion.
Reviewer 2 Report
Comments and Suggestions for Authors
The manuscript by Kajla et al. is a compact and comprehensive overview on microscopy & imaging analysis methods available for characterization of biological tissues at the nanoscale. The authors structured the overview into four sections, i.e. EM, LM, others like AFM and chemical mapping, as well as AI tools for segmentaion and image reconstruction. For each section, a conceptual and technical overview of various approaches is presented, along with examples from the literature and some practical considerations and limitations for each method.
Despite of the concise compilation, the review could and should be improved in the following three aspects:
1) Abstract: The authors claim that the review concludes with a dicussion on live-tissue nanoscopy and correlative light- and electrone microscopy (CLEM). However, the Conclusion section p.20 does not address a disussion on live-tissue nanoscopy at all, and only includes one very general statement in l. 749-51 on "… integrate multiple approaches into correlative pipelines…". Both aspects are important developments in nanoscale imaging modalities with significant benefits since the last 10-15 years, and the authors should expand their review by including a section/paragraph on CLEM & other correlative modes, e.g. AFM & RAMAN or other combinations.
2) Section 2 EM, p.3ff: Since ther main application focus of the manuscript is on biological tissues, the authors are asked to add a paragraph on ESEM modality.
3) Section 4 Mechanical and Chemical Nanoscale Imaging, p.14ff: In addition to SIMS, the authors may also include RAMAN spectroscopy, and in particular AFM/RAMAN combination as an example for local chemical analysis.
Author Response
General Comment: The manuscript by Kajla et al. is a compact and comprehensive overview on microscopy & imaging analysis methods available for characterization of biological tissues at the nanoscale. The authors structured the overview into four sections, i.e. EM, LM, others like AFM and chemical mapping, as well as AI tools for segmentaion and image reconstruction. For each section, a conceptual and technical overview of various approaches is presented, along with examples from the literature and some practical considerations and limitations for each method.
Response: Thank you for the positive assessment and for summarizing the structure of our manuscript. We appreciate the reviewer’s recognition of the comprehensive and conceptually organized overview across EM, LM, mechanical/chemical methods, and AI-based image analysis.
Despite of the concise compilation, the review could and should be improved in the following three aspects:
Comment 1 Abstract: The authors claim that the review concludes with a dicussion on live-tissue nanoscopy and correlative light- and electrone microscopy (CLEM). However, the Conclusion section p.20 does not address a disussion on live-tissue nanoscopy at all, and only includes one very general statement in l. 749-51 on "… integrate multiple approaches into correlative pipelines…". Both aspects are important developments in nanoscale imaging modalities with significant benefits since the last 10-15 years, and the authors should expand their review by including a section/paragraph on CLEM & other correlative modes, e.g. AFM & RAMAN or other combinations.
Response: Thank you for this insightful comment. We agree that both live-tissue nanoscopy and correlative light–electron microscopy (CLEM) represent important developments that were not sufficiently reflected in the original Conclusion.
Our modification to the manuscript: We have added a dedicated section discussing recent advances in AFM–Raman/TERS combinations and other correlative modalities. In addition, we introduced brief clarifying statements within the relevant sections to highlight how these correlative approaches bridge complementary strengths of different modalities whil discussing the Raman spectroscopy technique. This revision ensures that the manuscript’s conclusions accurately reflect the topics mentioned in the Abstract and emphasizes their significance in current nanoscale imaging workflows.
Our modification to the manuscript: We have added a dedicated paragraph describing correlating Raman spectroscopy with optical imaging modalities and electron microscopy.
"Biological applications of TERS have expanded significantly as instrumentation and probe stability have improved. Fig. 11c shows a representative example in which correlative AFM, confocal Raman, and TERS imaging are used to map the biochemical organization of a BxPC-3 human pancreatic cancer cell membrane [204]. ... Continued advances in probe design, drift-correction, multi-modal AFM-TERS integration, and machine-learning-assisted spectral analysis are expected to further expand the role of TERS in nanoscale tissue characterization, enabling researchers to link biochemical identity with structural context in complex biological systems. Furthermore, with the advancement in correlative imaging techniques, Raman imaging, fluorescence microscopy, and electron microscopy can complement each other and clarify morphological, structural, and chemical information of materials at the nanoscale [205,206].”
Comment 2 Section 2 EM, p.3ff: Since the main application focus of the manuscript is on biological tissues, the authors are asked to add a paragraph on ESEM modality.
Response: Thank you for this valuable suggestion. We agree that Environmental SEM (ESEM) is particularly relevant for imaging hydrated or sensitive biological tissues.
Our modification to the manuscript: We have added a dedicated paragraph in Section 2 describing the principles, advantages, and limitations of ESEM.
“Environmental scanning electron microscopy (ESEM) represents an extension of conventional SEM that enables imaging of biological tissues under partially hydrated, low-vacuum, or variable-pressure conditions. Unlike high-vacuum SEM, which requires extensive fixation, dehydration, and conductive coating, ESEM allows imaging of uncoated and minimally processed specimens by introducing water vapor or inert gas into the chamber to stabilize surface charge and prevent sample desiccation [60-62]. This capability is useful for soft tissues, cells, and extracellular matrix components that are easily distorted during drying steps. The secondary electron detector used in ESEM allows high-contrast imaging even at chamber pressures of several hundred Pa, which helps in the visualization of surface morphology, hydration-dependent features, and dynamic processes such as swelling, deformation, or phase transitions. In biological tissue research, ESEM has been used to observe hydrated cartilage, skin, plant tissues, biofilms, and wet cellular surfaces, providing micro- to nanoscale insights without conductive coating or high-vacuum artifacts [46,63-66]. While ESEM generally offers lower resolution than high-vacuum SEM, its ability to image delicate biological samples in near-native states makes it an essential complementary modality within the EM toolkit for nanoscale tissue characterization.”
Comment 3 Section 4 Mechanical and Chemical Nanoscale Imaging, p.14ff: In addition to SIMS, the authors may also include RAMAN spectroscopy, and in particular AFM/RAMAN combination as an example for local chemical analysis.
Response: Thank you for this helpful recommendation. We agree that Raman-based approaches, particularly AFM–Raman and tip-enhanced Raman spectroscopy (TERS), provide important complementary chemical information at the nanoscale.
Our modification to the manuscript: We have expanded Section 4 to include a dedicated paragraph on Raman spectroscopy in the context of tissue characterization, with specific emphasis on AFM–Raman/TERS correlative imaging. This new subsection
“4.3 Tip-enhanced Raman Spectroscopy
Tip-enhanced Raman spectroscopy (TERS) is an advanced near-field vibrational imaging technique that can be used for label-free, high-resolution chemical mapping of biological samples by overcoming the diffraction-limited spatial resolution of conventional Raman microscopy [194-201]. While far-field Raman can identify biomolecular signatures, its spatial resolution is typically limited to several hundred nanometers, restricting its ability to probe the nanoscale heterogeneity inherent to cell membranes, protein complexes, and subcellular structures. Fig. 11a illustrates the TERS principle, in which an excitation laser illuminates a sharp metallic AFM tip brought into nanometer proximity with the sample surface [202]. The confined optical near-field at the tip apex enhances Raman scattering from molecules directly beneath the tip, providing chemical sensitivity at the 10–20 nm scale which is an order-of-magnitude improvement over confocal Raman imaging. The optical performance of TERS strongly depends on illumination geometry, as shown in Fig. 11b [203]. Bottom-illumination offers high enhancement efficiency but is restricted to transparent substrates, whereas side- and top-illumination geometries accommodate opaque biological tissues and require long-working-distance objectives for stable excitation and collection.
Biological applications of TERS have expanded significantly as instrumentation and probe stability have improved. Fig. 11c shows a representative example in which correlative AFM, confocal Raman, and TERS imaging are used to map the biochemical organization of a BxPC-3 human pancreatic cancer cell membrane [204]. Confocal Raman images constructed from protein marker bands such as phenylalanine (1001 cm⁻¹) and histidine (1532 cm⁻¹) exhibit little spatial contrast due to diffraction-limited sampling. In contrast, TERS images can effectively identify the pronounced nanoscale variations, resolving chemically distinct domains separated by only tens of nanometers. Extracted spectra from high- and low-intensity TERS pixels further confirm localized enrichment of specific amino-acid signatures, underscoring the ability of TERS to detect molecular heterogeneity at the level of individual membrane components. These capabilities make TERS particularly valuable for investigating cell-membrane organization, protein aggregation, amyloid formation, bacterial envelope chemistry, and extracellular matrix remodeling. Continued advances in probe design, drift-correction, multi-modal AFM-TERS integration, and machine-learning-assisted spectral analysis are expected to further expand the role of TERS in nanoscale tissue characterization, enabling researchers to link biochemical identity with structural context in complex biological systems. Furthermore, with the advancement in correlative imaging techniques, Raman imaging, fluorescence microscopy, and electron microscopy can complement each other and clarify morphological, structural, and chemical information of materials at the nanoscale [205,206].”
We also added a new figure illustrating tip-based Raman imaging to enhance clarity and completeness.
Reviewer 3 Report
Comments and Suggestions for Authors
The review is well-written clear ordered and concise, and scans major imaging techniques suitable for biological tissues. I just wonder if other imaging techniques do not worth to be mentioned and compared with, such as light-sheet (little toxicity), two-photon (less agressive), hyperspectral (optical fingerprint), scanning tunneling microscopy, tip-enhanced Raman spectroscopy, ...
Author Response
Comment: The review is well-written clear ordered and concise, and scans major imaging techniques suitable for biological tissues. I just wonder if other imaging techniques do not worth to be mentioned and compared with, such as light-sheet (little toxicity), two-photon (less agressive), hyperspectral (optical fingerprint), scanning tunneling microscopy, tip-enhanced Raman spectroscopy, ...
Response: Thank you for the positive evaluation and for suggesting additional imaging techniques that could be considered. We appreciate the reviewer’s insight regarding modalities such as light-sheet microscopy, two-photon excitation, hyperspectral imaging, scanning tunneling microscopy, and tip-enhanced Raman spectroscopy (TERS).
Our modification to the manuscript: The aim of this review is to focus specifically on nanoscale characterization techniques, whereas modalities such as light-sheet, two-photon, and hyperspectral optical imaging typically operate under diffraction limit (>200 nm) and are therefore complementary rather than central to our scope. However, we agree that tip-enhanced Raman spectroscopy (TERS) is directly relevant to nanoscale chemical analysis, and we have now included a dedicated subsection and figure on TERS in the revised manuscript. This allows us to maintain a clear nanoscale focus while acknowledging the broader imaging landscape.
“4.3 Tip-enhanced Raman Spectroscopy
Tip-enhanced Raman spectroscopy (TERS) is an advanced near-field vibrational imaging technique that can be used for label-free, high-resolution chemical mapping of biological samples by overcoming the diffraction-limited spatial resolution of conventional Raman microscopy [194-201]. While far-field Raman can identify biomolecular signatures, its spatial resolution is typically limited to several hundred nanometers, restricting its ability to probe the nanoscale heterogeneity inherent to cell membranes, protein complexes, and subcellular structures. Fig. 11a illustrates the TERS principle, in which an excitation laser illuminates a sharp metallic AFM tip brought into nanometer proximity with the sample surface [202]. The confined optical near-field at the tip apex enhances Raman scattering from molecules directly beneath the tip, providing chemical sensitivity at the 10–20 nm scale which is an order-of-magnitude improvement over confocal Raman imaging. The optical performance of TERS strongly depends on illumination geometry, as shown in Fig. 11b [203]. Bottom-illumination offers high enhancement efficiency but is restricted to transparent substrates, whereas side- and top-illumination geometries accommodate opaque biological tissues and require long-working-distance objectives for stable excitation and collection.
Biological applications of TERS have expanded significantly as instrumentation and probe stability have improved. Fig. 11c shows a representative example in which correlative AFM, confocal Raman, and TERS imaging are used to map the biochemical organization of a BxPC-3 human pancreatic cancer cell membrane [204]. Confocal Raman images constructed from protein marker bands such as phenylalanine (1001 cm⁻¹) and histidine (1532 cm⁻¹) exhibit little spatial contrast due to diffraction-limited sampling. In contrast, TERS images can effectively identify the pronounced nanoscale variations, resolving chemically distinct domains separated by only tens of nanometers. Extracted spectra from high- and low-intensity TERS pixels further confirm localized enrichment of specific amino-acid signatures, underscoring the ability of TERS to detect molecular heterogeneity at the level of individual membrane components. These capabilities make TERS particularly valuable for investigating cell-membrane organization, protein aggregation, amyloid formation, bacterial envelope chemistry, and extracellular matrix remodeling. Continued advances in probe design, drift-correction, multi-modal AFM-TERS integration, and machine-learning-assisted spectral analysis are expected to further expand the role of TERS in nanoscale tissue characterization, enabling researchers to link biochemical identity with structural context in complex biological systems. Furthermore, with the advancement in correlative imaging techniques, Raman imaging, fluorescence microscopy, and electron microscopy can complement each other and clarify morphological, structural, and chemical information of materials at the nanoscale [205,206].”
We also added a new figure illustrating tip-based Raman imaging to enhance clarity and completeness.
Reviewer 4 Report
Comments and Suggestions for Authors
This is a beautifully well-written and informative summary of nanoscale imaging techniques. However, the figures are impossible to enjoy due to such a large amount of material presented in a small area. I suggest moving the schematics to supplemental data where those who want more information have those summary schematics to better understand the techniques and limit the figures in the main manuscript to the beautiful images. Perhaps even split the images into multiple figures so all the images can be well visualized.
Author Response
Comment: This is a beautifully well-written and informative summary of nanoscale imaging techniques. However, the figures are impossible to enjoy due to such a large amount of material presented in a small area. I suggest moving the schematics to supplemental data where those who want more information have those summary schematics to better understand the techniques and limit the figures in the main manuscript to the beautiful images. Perhaps even split the images into multiple figures so all the images can be well visualized.
Response: Thank you for encouraging feedback and for highlighting the issue of figure density. We appreciate the reviewer’s suggestion to reorganize the figures to improve readability.
Our modification to the manuscript: In the revised version, we have split all composite figures into multiple focused figures with shorter captions and placed each within the appropriate subsection. In the revised version, we have split all composite figures into multiple focused figures, each positioned within its relevant subsection and accompanied by much shorter, clearer captions. Specifically, the original Figure 2 (electron microscopy) was divided into three independent figures, Figure 3 (optical nanoscopy) into three figures, Figure 5 (AFM and nanoSIMS) into two figures, and Figure 6 into two figures. While we retained key schematics in the main text to improve understanding, we significantly reduced the complexity and ensured they no longer overcrowd the figures. This approach preserves accessibility for readers while enhancing overall readability and visual quality.
Round 2
Reviewer 2 Report
Comments and Suggestions for Authors
Dear authors, the revision of your nanoscale imaging review mansucript by Kajla et al. is now expanded and significantly improved over the original version. The reviewer acknowledges the constructive adding of new sections on ESEM and TERS/Raman-AFM to the review, as well as the refinement of the text that is now more coherent throughout, including the sections of AI-driven segmentation and reconstruction.